# Big Data Supported the Identification of Urban Land Efficiency in Eurasia by Indicator SDG 11.3.1

**Chaopeng Li** [1], **Guoyin Cai** [1,2,*] **and Mingyi Du** [1,2]

1   School of Geomatics and Urban Spatial Informatics, Beijing University of Civil Engineering and Architecture, Beijing 100044, China; 2108521519048@stu.bucea.edu.cn (C.L.); dumingyi@bucea.edu.cn (M.D.)
2   Beijing Advanced Innovation Center for Future Urban Design, Beijing University of Civil Engineering and Architecture, Beijing 100044, China
*   Correspondence: cgyin@bucea.edu.cn

**Abstract:** Indicator 11.3.1 of the UN Sustainable Development Goals (SDG 11.3.1) was designed to test land-use efficiency, which was defined as the ratio of the land consumption rate (LCR) to the population growth rate (PGR), namely, LCRPGR. This study calculates the PGRs, LCRs, and LCRPGRs for 333 cities from 1990–2000 and 391 cities from 2000–2015 in four geographical divisions in Eurasia according to the method given by UN metadata. The results indicate that Europe and Japan have the lowest PGR and LCR, indicating that this region's level of urbanization is the highest. South and Central Asia have the lowest values of LCRPGR, indicating relatively lower urban land supply during the measurement periods. Compared with the mean LCRPGR in a region, the average values from SDG 11.3.1 by different types of cities in a region can have more guiding significance for urban sustainable development. While paying attention to the urban land-use efficiency of mega and extra-large cities, more attention should be paid to the coordination relationship between urban land supply and population growth in large, medium, and small cities. Additionally, the method from UN metadata works well for most urban expansion cities but is not suitable for cities with small changes in urban populations.

**Keywords:** urban land use efficiency; SDG 11.3.1; Eurasia; big Earth data

## 1. Introduction

With the intensification of global urbanization, the proportion of the global urban population has increased from less than 30% in 1950 to 55% in 2018, and this is expected to increase to 68% by 2050 [1]. Rapid urban expansion occupies arable and other land resources; therefore, the continuous reduction of farmland, green spaces, or other water-pervious surfaces has emerged as an important issue, especially in Africa, Asia, and Latin America over the last decades [2–4]. At the same time, the imbalance between urban land supply and population growth reduces the efficiency of land use, which negatively affects the sustainable development of a city. This has been proven by a project jointly completed by the African Union, African Development Bank, and Economic Commission on 120 cities. Their results showed that urban land cover grew by more than three times the increase in urban population [5]. This rate of urban expansion has hindered urban sustainable development.

In 2015, the United Nations (UN) issued the Sustainable Development Goals (SDGs). The 2030 Agenda contains 17 SDGs, with 169 related targets and 232 indicators addressing the social, economic, and environmental dimensions of development [6]. The centrality of cities in the global sustainability challenge is widely acknowledged [7–9]. One of the UN's SDGs is specifically for urban sustainable development, that is, SDG 11. SDG 11 aims to 'make cities and human settlements inclusive, safe, resilient and sustainable' [10]. There are 10 targets and 15 indicators in SDG 11, of which SDG 11.3.1, the ratio (LCRPGR) of the land consumption rate (LCR) to the population growth rate (PGR), can be used to

identify the urban land-use efficiency (ULUE) [11,12]. ULUE reflects the ability to promote the coordinated development of the urban society, economy, and environment. In addition to social and economic output, ULUE also attaches importance to ecological benefits, emphasizing the balance between land development and environmental protection [13]. It is an indicator of achievable goals and resources that can be consumed [14]. From a socioeconomic and ecological perspective, the rational and effective use of land is essential for sustainable development. Understanding ULUE in a specific area can help the sustainable development of land use in the area [15]. On the contrary, inefficient urban land use poses a severe challenge to the sustainable development of cities. Therefore, understanding ULUE is essential to design an appropriate land policy or to fill in the gaps of existing policies.

Many indicators, such as the rate of urban boundary expansion [16], the scale of idle land [17], urban densification [18], and localized urban sustainability measurement factors [19,20], can be used to identify the ULUE. The UN employed the ratio of build-up expansion to the population size for a specific time period. The Sustainable Development Report released by UN-Habitat in 2017 showed that from 2000 to 2015, the rate of urban land expansion exceeded the rate of urban population growth in all regions of the world; that is, LCRPGR was greater than 1. The mean LCRPGR value in 120 cities around the world increased from 1.22 in 1990–2000 to 1.28 in 2000–2015 [21,22]. In addition to international organizations, the academic community has also carried out relevant research on indicator SDG 11.3.1. The Earth observation data and derived datasets have proven to be feasible for monitoring land degradation or water-related ecosystem changes according to the UN SDG framework at various geographical scales [23–25]. The Global Human Settlement Layer (GHSL) provides open data on the spatial distribution of built-up areas, population, and settlement typologies for the years 1975, 1990, 2000, and 2015 (http://sedac.ciesin.columbia.edu/data/collection/gpw-v4) [26]. Melchiorri et al. (2019) discussed the availability of the GHSL in the calculation of SDG 11.3.1 indicators, indicating that GHSL can be used in the calculation of SDG 11.3.1 on a global scale [27]. The results by Wang et al. (2020) in China showed that among the megacities with a population of 5–10 million in China, more than 95% of cities have a significant increase in urban built-up areas by employing national-scale open data sources. The mean LCRPGR increased from 1.69 in 1990–2000 to 1.78 in 2000–2010. This means that the LCR is 1.78 times the PGR from 2000 to 2010 in China [28]. A study in Addis Ababa, Ethiopia, by Koroso et al. (2020) indicated that the LCRPGR of Addis Ababa was 1.02 from 2005 to 2019, while the overall LCRPGR values of two suburban areas of Bole and Akaki-Kaliti were 3.16 and 3.62, respectively, between 2007 and 2019, indicating low ULUE for both the cities and suburban areas [18].

Current studies mostly carry out the calculation of SDG 11.3.1 from a global or national perspective using accessible open data which may have uncertainty problems. This study employs spatial and statistical data to compute indicator SDG 11.3.1 in Eurasia to identify the regional land-use efficiency using the SDG 11.3.1 model given by the UN metadata. With the help of big Earth data and the UN's population data, this work calculates the LCRPGR values and examines land-use efficiencies in measurement periods of 1990–2000 and 2000–2015 in Eurasia using SDG 11.3.1. The remainder of this paper is organized as follows: Section 2 introduces the methods, including the study area, data sources, and data processing, as well as the method to calculate indicator SDG 11.3.1; Section 3 details the results and analysis; Section 4 provides a discussion; and Section 5 concludes the paper.

## 2. Materials and Methods

In Section 2.1, we briefly introduce the study site and city samples and distributions in the study area. In Section 2.2, we present the data sources and processing procedures, and Section 2.3 describes the method that was applied in this study to estimate SDG 11.3.1 according to the UN metadata.

### 2.1. Study Area

Eurasia is composed primarily of landmasses that are conventionally divided into the two continents of Asia and Europe [29]. The study region was chosen to approximate the extent of most parts of Eurasia, with coordinates of 10.225° S, 13.25° W, and 69.562° N, 144.323° E, which have undergone severe dynamic changes in land use/land cover in the past 40 years [30]. Based on the global geographical division by UN and the city numbers employed in each region, four geographic regions, namely Europe and Japan, South and Central Asia, Southeast Asia, and Western Asia, were classified. Figure 1 shows the study regions and cities employed in this study.

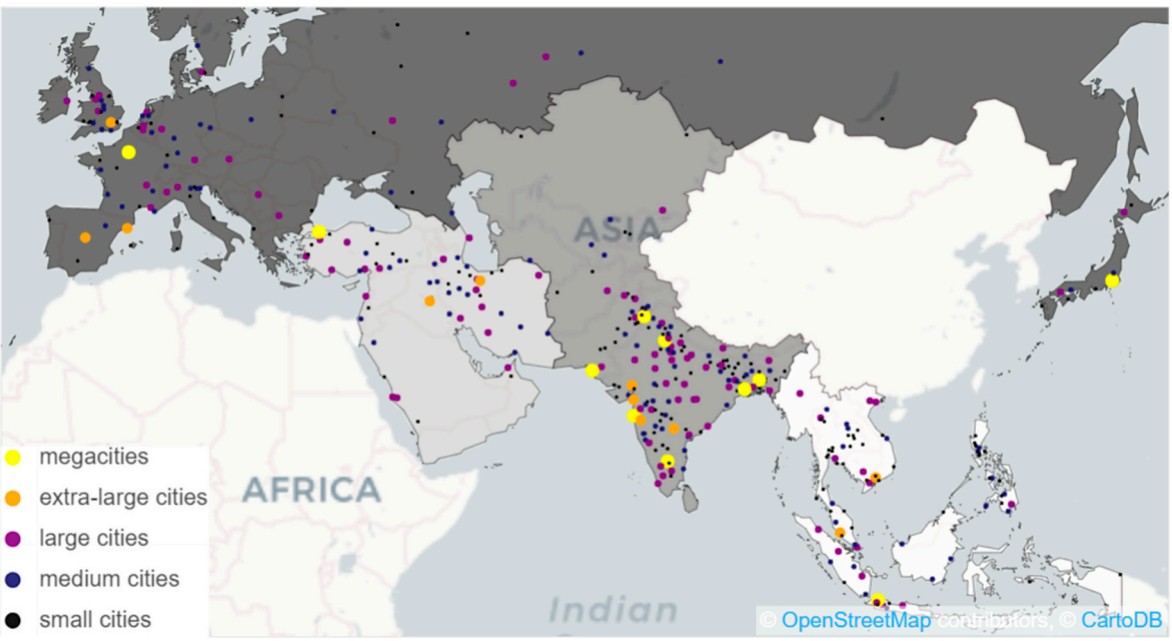

**Figure 1.** Cities in the four geographic regions of the study area.

### 2.2. Data and Data Processing

Previous works have proven that the combination of satellite imagery, social media data, and population data are useful as a proxy for interpreting urban sprawl and its impact on urban sustainable development [31,32]. The data employed in this paper are mainly the basic geographic data, urban impervious surface data, and urban population data in Eurasia. The urban population data came from the global statistical population data released by the UN in 2018. There were 421 cities and 534 cities with more than 300,000 people in 2000 and 2015, respectively. Data of these cities was evaluated during 1990–2000 and 2000–2015, respectively. The urban impervious surface data comes from the 30 m high-resolution urban impervious surface dataset of 1990, 2000, and 2015 from the Big Data Platform of the Chinese Academy of Sciences [33]. This dataset was produced mainly using Sentinel-1A and Sentinel-2 imagery with an overall accuracy greater than 88.00%. First, the potential urban land (PUL) was identified through logical operations on yearly mean and standard deviation composites from Sentinel-1 images. Second, vegetation, water, and mountain pixels were masked by applying thresholds for the yearly normalised difference vegetation index maximum, modified normalised difference water index mean, and slope images, respectively. Third, a region-specific threshold was assigned to the PUL to extract the target urban land (TUL). Finally, a majority filter with a 3 × 3 window was applied to the TUL to obtain the impervious surface data [34]. According to the rule recommended by the UN on the definition of urban areas, taking into account size and distance [35], urban areas of urban land of 20 or more hectares that are less than 200 m

apart were linked to form a continuous urban area. The impervious surface data were converted into urban built-up area data according to this definition of urban areas.

### 2.3. Methods

The method for calculating indicator SDG 11.3.1 is presented in the SDG indicators Metadata Repository managed by UNDESA (https://unstats.un.org/sdgs/metadata). Indicator SDG 11.3.1 is defined as the ratio of the LCR to the PGR. It is mainly used to identify land-use efficiency with the rapid development of urbanization around the world [36].

The PGR is the population growth in a city over a certain period. It reflects the number of births and deaths and the number of people moving in and out of a city in a certain period. According to the metadata, the calculation of the PGR is expressed as:

$$\text{PGR} = \frac{\text{Ln}(Pop_{t+n}/Pop_t)}{y} \tag{1}$$

where Ln is the natural logarithm, $Pop_t$ is the total population within the city in the initial year, $Pop_{t+n}$ is the total population within the city in the final year, and y is the number of years between the two measurement periods.

The LCR is the expansion of urban built-up areas over a period of time. It measures the percentage of current total urban land that was newly developed in a given spatial unit over a time span. The calculation of LCR is expressed as:

$$\text{LCR} = \frac{\text{Ln}(Urb_{t+n}/Urb_t)}{y} \tag{2}$$

where $Urb_t$ and $Urb_{t+n}$ are the total areal extent of the urban agglomeration in km$^2$ for the initial and final year, respectively; Ln and y have the same meanings as in Equation (1).

Both PGR and LCR adopt the exponential growth model. For instance, the population in the final year that grows exponentially over time is modelled by:

$$Pop_{t+n} = Pop_t \cdot e^{PGR \cdot y}$$

$$\frac{Pop_{t+n}}{Pop_t} = e^{PGR \cdot y}$$

$$Ln\left(\frac{Pop_{t+n}}{Pop_t}\right) = Ln(e^{PGR \cdot y})$$

$$PGR = Ln\left(\frac{Pop_{t+n}}{Pop_t}\right)/y$$

The indicator SDG 11.3.1 is defined as the ratio of LCR to PGR, namely, LCRPGR. It can be used to quantify sustainable land use in the face of urban expansion pressure. The estimate of the LCRPGR is expressed as:

$$LCRPGR = \frac{LCR}{PGR} = \frac{Ln(Urb_{t+n}/Urb_t)}{Ln(Pop_{t+n}/Pop_t)} \tag{3}$$

Equation (3) can be expressed as:

$$\text{Ln}\left(\frac{\text{Urb}_{t+n}}{\text{Urb}_t}\right) = \text{LCRPGR} \times \text{Ln}\left(\frac{\text{Pop}_{t+n}}{\text{Pop}_t}\right) \tag{4}$$

i.e.,

$$\frac{\text{Urb}_{t+n}}{\text{Urb}_t} = \left(\frac{\text{Pop}_{t+n}}{\text{Pop}_t}\right)^{\text{LCRPGR}} \tag{5}$$

Equation (5) shows that the method adopted by SDG indicators Metadata Repository is based on the assumption that the areal extent ratio of the urban agglomeration fits the power-law of the urban population ratio during a measurement period.

## 3. Results

The values of SDG 11.3.1 for Eurasian cities in the years 1990, 2000, and 2015 were calculated according to the LCRPGR model given in Equation (3). Through the analysis of the LCRPGR results, it was found that the LCRPGR in the calculation by the model provided by UN metadata has large positive values or small negative values. Some of these abnormal values are listed in Table 1.

The large positive or negative values in Table 1 are primarily caused by small changes in population growth over a period. A very small change in population causes the denominator in the LCRPGR calculation model to be very small either positively or negatively, while urban areas are expanded during the measurement period, which increases the numerator. As a result, the LCRPGR value becomes extremely positive or negative. If these abnormal values are involved in the regional assessment of indicator SDG 11.3.1, they can introduce large errors to the comprehensive evaluation of the indicator in a region or area and can greatly impact the subsequent urban sustainability analysis. According to UN-Habitat statistics, in some unplanned countries and regions in Africa, the LCR can be 3–5 times the PGR (UN habitat, 2018). Therefore, this study eliminates LCRPGR outliers less than −5 and greater than 5 for subsequent statistics. Based on the urban population division criteria [26], all cities were divided into five sizes based on the urban population: (1) megacities: >10,000,000; (2) extra-large cities: 5,000,000–10,000,000; (3) large cities: 1,000,000–5,000,000; (4) medium cities: 500,000–1,000,000; and (5) small cities: <500,000. In each area with the same urban size for every geographical region, three types of urban expansion were categorized (Table 2) based on the ratio of the LCR to PGR: $1 \leq$ LCRPGR $\leq 5$, $0 \leq$ LCRPGR $< 1$, and $-5 \leq$ LCRPGR $<0$.

**Table 1.** Samples of the abnormal values of the ratio of the land consumption rate (LCR) to the population growth rate (PGR), known as LCRPGR, from 1990 to 2000.

| Urban Agglomeration | Pop1990 | Pop2000 | Urb1990 | Urb2000 | LCR | PGR | LCRPGR |
|---|---|---|---|---|---|---|---|
| Mannheim | 308.178 | 307.229 | 54.1404 | 64.3193 | 0.0172 | −0.0003 | −55.86 |
| Bydgoszcz | 375.040 | 375.744 | 20.2907 | 27.8285 | 0.0316 | 0.0002 | 168.45 |
| Gdańsk | 458.139 | 461.403 | 27.0025 | 34.0577 | 0.0232 | 0.0007 | 32.70 |
| Astrakhan | 505.451 | 506.224 | 61.8521 | 66.0807 | 0.0066 | 0.0002 | 43.27 |
| Barnaul | 597.040 | 602.299 | 38.3597 | 57.4079 | 0.0403 | 0.0009 | 45.97 |
| Izhevsk | 632.842 | 632.237 | 23.9150 | 40.1962 | 0.0519 | −0.0001 | −542.90 |
| Kazan | 1092.396 | 1095.811 | 45.2132 | 57.6676 | 0.0243 | 0.0003 | 77.95 |
| Novosibirsk | 1430.038 | 1426.421 | 66.2444 | 104.8245 | 0.0459 | −0.0003 | −181.22 |
| Omsk | 1143.813 | 1135.729 | 30.2043 | 44.1858 | 0.0380 | −0.0007 | −53.64 |
| Orel | 337.937 | 334.399 | 14.1538 | 24.4874 | 0.0548 | −0.0011 | −52.09 |
| Volgograd | 999.426 | 1010.310 | 41.9957 | 70.8609 | 0.0523 | 0.0011 | 48.30 |
| Cordoba | 308.703 | 308.373 | 14.4439 | 18.9319 | 0.0271 | −0.0001 | −252.98 |

Note: Pop1990 and Pop2000 refer to the urban population with a unit of thousand people, while Urb1990 and Urb2000 refer to the urban built-up area with unit of km$^2$.

**Table 2.** Urban expansion types based on LCRPGR values.

| Urban Expansion Type | LCRPGR Data Range | Meanings |
|---|---|---|
| Rapid urban land growth | [1–5] | Urban land consumption exceeds population growth |
| Rapid urban population growth | (0–1) | Urban population growth exceeds land consumption |
| Urban shrinking | [−5–0) | Urban population declining or urban land shrinking |

After removing the abnormal values of LCRPGR greater than 5 or less than −5, 333 of the 421 cities from 1990–2000 and 391 of the 534 cities from 2000–2015 were selected to evaluate land-use efficiency. To obtain a clear comparison in the study area, the mean LCRP-GRs in each urban expansion type, urban size, and geographical region were calculated separately and listed in Table 3.

**Table 3.** Statistical LCRPGR values for 1990–2000 and 2000–2015 in Eurasia.

| Geo. Divisions | City Size | LCRPGR 1990–2000 | | | | LCRPGR 2000–2015 | | | |
|---|---|---|---|---|---|---|---|---|---|
| | | Mean | Data Range | Mean | City Number | Mean | Data Range | Mean | City Number |
| Europe and Japan | Mega | 1.73 | [1–5] | 1.73 | 1 | 1.19 | [1–5] | 2.12 | 1 |
| | | | (0–1) | – | – | | (0–1) | 0.73 | 2 |
| | Extra-large | 1.10 | [1–5] | 1.70 | 2 | 1.08 | [1–5] | 1.49 | 2 |
| | | | (0–1) | 0.70 | 3 | | (0–1) | 0.25 | 1 |
| | Large | 0.75 | [1–5] | 2.42 | 20 | 2.36 | [1–5] | 2.53 | 26 |
| | | | (0–1) | 0.79 | 4 | | (0–1) | 0.92 | 3 |
| | | | [−5–0] | −2.62 | 10 | | [−5–0] | – | – |
| | Medium | 0.76 | [1–5] | 2.50 | 20 | 1.82 | [1–5] | 2.46 | 31 |
| | | | (0–1) | 0.47 | 12 | | (0–1) | 0.67 | 8 |
| | | | [−5–0] | −2.53 | 9 | | [−5–0] | −3.47 | 2 |
| | Small | 0.37 | [1–5] | 2.48 | 26 | 2.05 | [1–5] | 2.50 | 39 |
| | | | (0–1) | 0.43 | 23 | | (0–1) | 0.68 | 5 |
| | | | [−5–0] | −2.58 | 19 | | [−5–0] | −1.51 | 3 |
| | Average | 0.61 | [1–5] | 2.44 | 69 | 2.00 | [1–5] | 2.48 | 99 |
| | | | (0–1) | 0.49 | 42 | | (0–1) | 0.70 | 19 |
| | | | [−5–0] | −2.58 | 38 | | [−5–0] | −2.29 | 5 |
| South and Central Asia | Mega | 0.87 | [1–5] | 1.04 | 2 | 1.35 | [1–5] | 2.21 | 3 |
| | | | (0–1) | 0.71 | 2 | | (0–1) | 0.70 | 4 |
| | Extra-large | 1.33 | [1–5] | 1.50 | 3 | 1.52 | [1–5] | 3.27 | 2 |
| | | | (0–1) | 0.82 | 1 | | (0–1) | 0.74 | 2 |
| | Large | 1.42 | [1–5] | 1.91 | 19 | 1.10 | [1–5] | 1.74 | 21 |
| | | | (0–1) | 0.71 | 13 | | (0–1) | 0.70 | 23 |
| | | | [−5–0] | – | – | | [−5–0] | −3.39 | 1 |
| | Medium | 1.26 | [1–5] | 2.20 | 15 | 1.52 | [1–5] | 2.03 | 27 |
| | | | (0–1) | 0.63 | 15 | | (0–1) | 0.54 | 14 |
| | | | [−5–0] | −1.79 | 1 | | [−5–0] | – | – |
| | Small | 1.54 | [1–5] | 2.21 | 25 | 1.71 | [1–5] | 2.31 | 40 |
| | | | (0–1) | 0.61 | 18 | | (0–1) | 0.57 | 21 |
| | | | [−5–0] | – | – | | [−5–0] | – | – |
| | Average | 1.41 | [1–5] | 2.05 | 64 | 1.47 | [1–5] | 2.09 | 93 |
| | | | (0–1) | 0.65 | 49 | | (0–1) | 0.63 | 64 |
| | | | [−5–0] | −1.79 | 1 | | [−5–0] | −3.39 | 1 |
| Southeast Asia | Mega | 3.74 | [1–5] | 3.74 | 1 | 1.73 | [1–5] | 1.73 | 1 |
| | Extra-large | – | – | – | – | 1.10 | [1–5] | 1.10 | 2 |
| | Large | 1.87 | [1–5] | 2.01 | 7 | 1.62 | [1–5] | 2.21 | 9 |
| | | | (0–1) | 0.89 | 1 | | (0–1) | 0.74 | 6 |
| | | | [−5–0] | – | – | | [−5–0] | – | – |
| | Medium | 2.22 | [1–5] | 2.65 | 11 | 1.59 | [1–5] | 1.81 | 17 |
| | | | (0–1) | 0.64 | 3 | | (0–1) | 0.64 | 4 |
| | | | [−5–0] | – | – | | [−5–0] | – | – |
| | Small | 1.43 | [1–5] | 1.13 | 10 | 1.14 | [1–5] | 1.95 | 20 |
| | | | (0–1) | 0.55 | 8 | | (0–1) | 0.74 | 16 |
| | | | [−5–0] | – | – | | [−5–0] | −0.02 | 1 |
| | Average | 1.84 | [1–5] | 2.36 | 29 | 1.48 | [1–5] | 1.91 | 49 |
| | | | (0–1) | 0.60 | 12 | | (0–1) | 0.73 | 26 |
| | | | [−5–0] | – | – | | [−5–0] | −0.02 | 1 |

**Table 3.** *Cont.*

| Geo. Divisions | City Size | LCRPGR 1990–2000 | | | | LCRPGR 2000–2015 | | | |
|---|---|---|---|---|---|---|---|---|---|
| | | Mean | Data Range | Mean | City Number | Mean | Data Range | Mean | City Number |
| Western Asia | Mega | – | – | – | – | – | – | – | – |
| | Extra-large | 0.97 | [1–5] | 1.38 | 1 | 1.39 | [1–5] | 1.39 | 2 |
| | | | (0–1) | 0.55 | 1 | | (0–1) | – | – |
| | Large | 1.53 | [1–5] | 2.55 | 5 | 0.62 | [1–5] | 2.07 | 6 |
| | | | (0–1) | 0.87 | 1 | | (0–1) | 0.73 | 3 |
| | | | [−5–0] | −2.92 | 1 | | [−5–0] | – | – |
| | Medium | 1.91 | [1–5] | 2.77 | 4 | 1.98 | [1–5] | 2.21 | 12 |
| | | | (0–1) | 0.76 | 3 | | (0–1) | 0.59 | 2 |
| | Small | 2.03 | [1–5] | 2.13 | 12 | 1.96 | [1–5] | 2.32 | 7 |
| | | | (0–1) | 0.75 | 1 | | (0–1) | 0.72 | 2 |
| | Average | 1.71 | [1–5] | 2.62 | 21 | 1.84 | [1–5] | 2.14 | 27 |
| | | | (0–1) | 0.71 | 7 | | (0–1) | 0.68 | 7 |
| | | | [−5–0] | −2.92 | 1 | | [−5–0] | – | – |

*3.1. Land-Use Efficiencies in Different Geographical Divisions in Eurasia*

Statistically, the mean value reflects the average dataset level. The analysis in this section is based on the mean values on the four geographical divisions from the data which were collected in this work. The mean PGR, LCR, and LCRPGR values within a data range of [−5–5] from 1990 to 2000 and 2000 to 2015 in the four geographical regions are presented in Figure 2. From a numerical point-of-view, the PGR is the lowest among the PGR, LCR, and LCRPGR. Compared to other regions, the PGR in Europe and Japan is much lower than that in other regions. From the period of 1990–2000 to 2000–2015, there is an increasing trend in the PGR in the Europe and Japan region, a decreasing trend in the regions of Southeast Asia and Central and South Asia, and a consistent trend in Western Asia. With the change in population growth, the LCR presents nearly the same changing trend as the PGR change in the rate values and trend. This means that urban land consumption maintains the same changing trend as the PGR. Both the LCR and PGR are the smallest in the Europe and Japan region, indicating a relatively high level of urbanization.

LCRPGR presents different changing conditions compared to LCR or PGR. LCRPGR mainly reflects the status of urban land provision as urban population growth increases during the process of urbanization. The lower LCRPGR value of 0.61 in Europe and Japan during 1990–2000 indicates that urban land provision is lower than the population growth, making the urban land denser. While other LCRPGR values greater than 1 indicate that the urban land consumption ratio is larger than the PGR. Compared with the world average LCRPGR value of 1.22 and 1.28 from the periods of 1990–2000 and 2000–2015 given by the Sustainable Development Goals Report 2017, except for the LCRPGR in 1990–2000 in Europe and Japan, all other regions do not follow the phenomenon in which the urban land consumption is greater than the world average as the urban population increases. This is especially true in the southern and central Asia regions, which have the densest urban land use in the measurement periods of 1990–2000 and 2000–2015, with nearly the same LCRPGR value. In the regions of Europe and Japan for 2000–2015, the largest LCRPGR values indicate sufficient urban land supply. The mean LCRPGR values can reflect the average land-use efficiency of an entire region. The difference among cities can be represented by standard errors in each geographical division (Figure 3). This shows the difference between LCRPGR values of individual cities and the average among cities in each region, reflecting the unbalanced ULUE among cities in each region. The largest difference existed in the Europe and Japan region that included more than 170 cities with a maximum LCRPGR of 4.87 and a minimum of −4.84 during 1990–2000. The large variations among LCRPGR values may have been caused by the large number of cities in the region covering a large area from west to east.

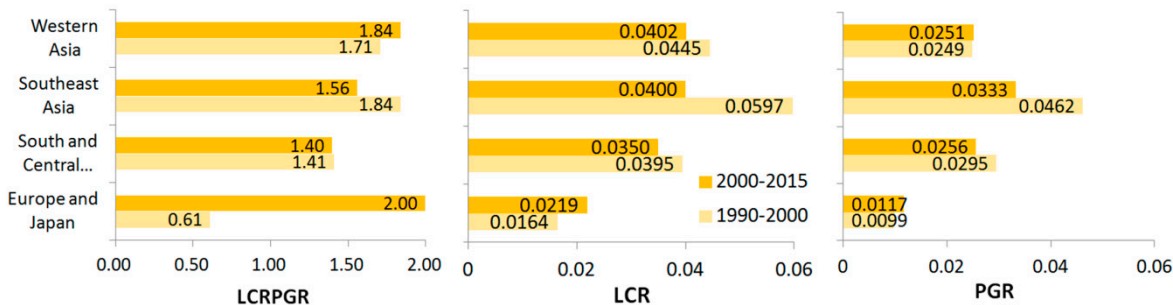

**Figure 2.** LCRPGR, LCR, and PGR values during 1990–2000 and 2000–2015 in different geographical regions of Eurasia.

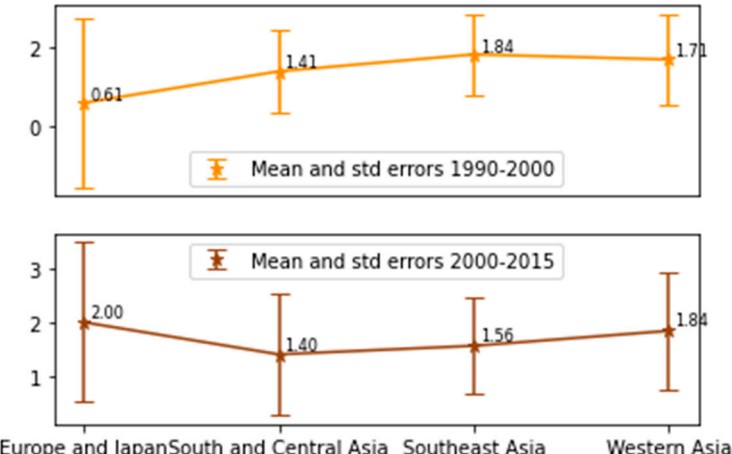

**Figure 3.** LCRPGR means and standard errors in each geographical division for the periods of 1990–2000 and 2000–2015.

### 3.2. Land-Use Efficiencies in Rapid Urban Land Growth Cities with Different Sizes in Eurasia

For a detailed analysis of land-use efficiencies in different types of urban expansion cities, the mean LCPGR values for rapid urban land growth cities with different urban sizes in the four geographical divisions are presented in Figure 4. Compared with the LCRGPR value of 1.22 for 1990–2000 and 1.28 for 2000–2015 given by the UN's 2017 report, the LCRPGRs of different-sized cities in most regions are relatively larger, indicating lower land-use efficiency in cities in Eurasia than that of the world average. Although Table 3 shows that there are only a few mega and extra-large cities in each geographical region, the mean LCRPGR is easily impacted by one or two cities, which makes the mean LCRPGR values in mega or extra-large cities statistically less significant. Figure 3 shows that the phenomenon of urban land expansion in large, medium, and small cities is more severe than that in extra-large and large megacities for most of the geographical regions in the measurement periods. This indicates that the urban lands in mega and extra-large cities are denser than those in large, medium, and small cities, indicating relatively higher urban land efficiency in mega and extra-large cities. This means that the land-use efficiency has increased as the process of urbanization has intensified.



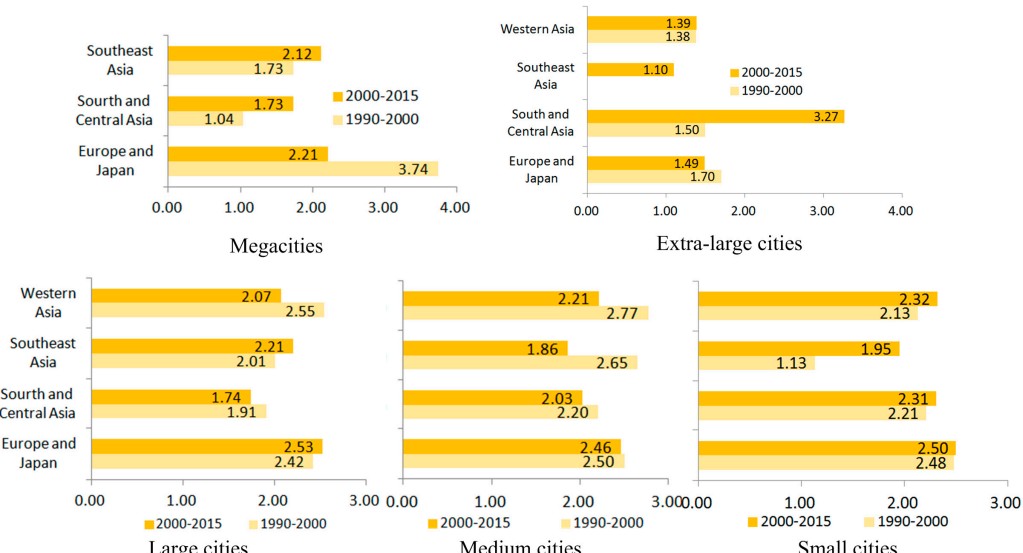

**Figure 4.** LCRPGR values for rapid urban land growth cities with different sizes for 1990–2000 and 2000–2015 in four geographical regions in Eurasia.

### 3.3. Land-Use Efficiencies in Rapid Urban Population Growth Cities with Different Sizes in Eurasia

In cities with rapid urban population growth, the LCR is lower than the urban PGR. This means that there is a lack of urban land provision as the population grows. A lower LCRPGR value indicates lower provision of urban land. Figure 5 presents the LCRPGR values for 1990–2000 and 2000–2015 in the four geographical regions with different urban sizes. A phenomenon in which the PGR is two to four times larger than the urban LCR is seen in extra-large, large, and small cities in Europe and Japan; while in other regions, the PGR is less than two times the urban LCR. Generally, only a small number of mega or extra-large cities belong to the rapid population growth type. They are concentrated in South and Central Asia, Europe, and Japan. Most cities with rapid population growth are large, medium, and small cities.

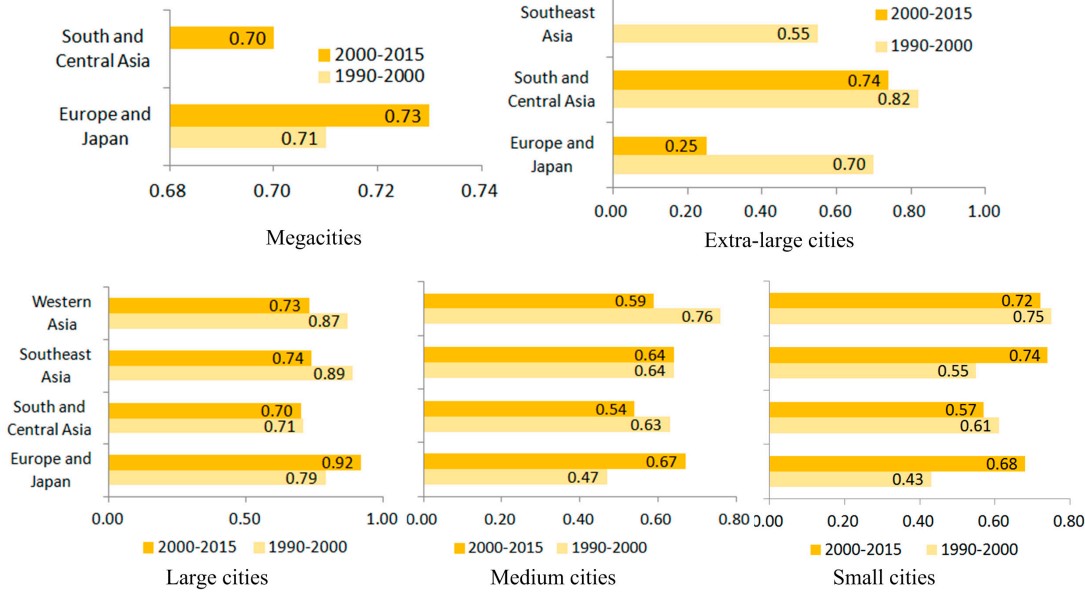

**Figure 5.** LCRPGR values for rapid population growth cities with different sizes during 1990–2000 and 2000–2015 in four geographical regions in Eurasia.

## 4. Discussion

SDG 11.3.1 was designed to analyze land-use efficiency as global urbanization threatens urban sustainable development. Earth Big Data provides powerful data support for the evaluation of land-use efficiency from a global or regional perspective. This study uses global impervious surface data to extract information on urban built-up areas and combines population data to calculate and analyze indicator SDG 11.3.1 of more than 300 cities in Eurasia. Compared with the calculation results of more than 120 cities on a global scale in the UN report, this study uses a large number of cities and concentrates on the areas in Eurasia. Compared with the world average of 1.28 for LCRPGR in the '2017 Sustainable Development Goals Report', except for the LCRPGR of Europe and Japan during 1990–2000, the LCRPGR of all other regions was higher than 1.28 in both measurement periods, indicating that the ratio of the LCR to the PGR in cities in Eurasia was higher than the world average. Among them, areas with the highest urban land-use density are South and Central Asia; their LCRPGR values hardly changed during 1990–2000 and 2000–2015, indicating that the relationship between urban land supply and population growth is relatively stable. The region with the greatest change is the Europe and Japan region, which increased from a value of 0.61 in 1990–2000 to a value of 2 in 2000–2015, indicating that the urban land supply situation has changed significantly. This may be due to two reasons. First, the average LCRPGR value given by UN SDG report 2017 was calculated primarily by statistical data from 120 cities worldwide, a different data source and different city types and distributions as those of this paper. Second, the different sized cities with different urban expansion types were all used in a geographical region to calculate the average LCRPGR, which mainly presents the regional average level of land-use efficiency without focusing on cities with a severe urban expansion phenomenon.

To distinguish land-use efficiency in different types of cities, this study analyses land-use efficiency in rapid urban land expansion and rapid urban population growth cities with different urban scales. Among the cities with rapid land growth, the LCRPGR of cities with different urban sizes in most regions is higher than the value from the 2017 UN report. Only small cities and megacities in Southeast Asia during 1990–2000 and extra-large cities in South and Central Asia during 2000–2015 have LCRPGR values below the global average. In cities with rapid population growth, the LCRPGR is less than the average given by the UN. This indicates that the calculation results by the UN indicator SDG 11.3.1 model are different for different types of cities. Therefore, the calculation of LCRPGR for different types of cities is more meaningful for the sustainability evaluation of cities. In contrast to mega or extra-large cities, large, medium, and small cities have more problems in terms of land-use efficiency and rapid population growth. Therefore, policies on urban land-use and urban populations should pay more attention to large, medium, and small cities.

It should be noted that there are many abnormal LCRPGR values. If normal LCRPGR values are defined in the range of −5 to 5, there are 88 and 143 cities in the periods of 1990–2000 and 2000–2015, accounting for 21% and 27%, respectively, with abnormal LCRPGR values. These abnormal LCRPGR values are primarily caused by small changes in population growth over a period. The calculation method for indicator SDG 11.3.1 given by UN metadata has the format of a logarithmic ratio between LCR and PGR. The computation, with a lower PGR rate as the denominator and a larger LCR rate as the numerator, makes the ratio, that is, LCRPGR, have a large positive or negative value. If these abnormal values are used to assess the ULUE, it will inevitably reduce the reliability of the results. Thus, work should be performed on the calculation model for SDG 11.3.1.

## 5. Conclusions

Earth Big Data provides a new approach to detect and analyze regional or global progress in social, economic, and ecological development. SDG indicator 11.3.1 was designed to analyze land-use efficiency as the urban population grows. In this study, high-resolution impervious surface data and population data were collected to calculate indicator

SDG 11.3.1, to analyze the land-use efficiency for more than 300 cities in Eurasia during 1990–2000 and 2000–2015. The conclusions drawn from this work are as follows:

In the four geographic regions of Eurasia, the changes in the LCR and PGR have kept pace from 1990–2000 to 2000–2015. Europe and Japan have the lowest PGR and LCR, indicating that the level of urbanization in this region is the highest. Compared with the global average LCRPGR value given by the 2017 UN report, except for Europe and Japan during 1990–2000, the land-use efficiency in other regions during 1990–2000 and 2000–2015 was lower than the global average. Central Asia and South Asia have the lowest values of LCRPGR, nearly without change during the two measurement periods, indicating that the urban land supply in this region is low and that the relationship between land consumption and population growth is relatively stable. The largest changes occurred in Europe and Japan, where the LCRPGR increased from 0.61 in 1990–2000 to 2.0 in 2000–2015, indicating that with population growth, the urban land supply has become more abundant.

Among cities with rapid land expansion, the LCRPGR values of most cities of different sizes are greater than 1.22 or 1.28; whereas for cities with rapid population growth, the LCRPGRs for cities of different sizes are less than 1. The average value of LCRPGR for all cities can characterize the average land-use efficiency of cities in a region. However, averaging eliminates the differences between different types of cities. Therefore, the calculation results of different types of cities can have more guiding significance for their development. Compared with mega or extra-large cities, large, medium, and small cities have more severe problems with inefficient land use and excessive population growth. Therefore, while focusing on the coordination between population growth and urban land consumption in mega or extra-large cities, more attention should be paid to the land-use efficiency of large, medium, and small cities.

The results from this work were obtained by controlling the LCRPGR values between −5 and +5. More than 20% of cities were excluded from this analysis due to abnormal LCRPGR values. The reason for these numerical anomalies is mainly due to the logarithmic function ratio in the calculation model of indicator SDG 11.3.1. Therefore, further research should focus on the optimization of the indicator SDG 11.3.1 model with the support of big Earth data.

**Author Contributions:** Conceptualization, Guoyin Cai; Data curation, Chaopeng Li; Formal analysis, Guoyin Cai; Funding acquisition, Guoyin Cai; Investigation, Chaopeng Li; Project administration, Mingyi Du; Resources, Mingyi Du; Supervision, Mingyi Du. All authors have read and agreed to the published version of the manuscript.

**Funding:** This work is financially supported by Strategic Priority Research Program of the Chinese Academy of Sciences (grant number XDA19030104), and by Beijing Advanced Innovation Center for Future Urban Design (No. UDC2018030611) and by National Key Research and Development Program of China (No. 2017YFB0503900-4-3).

**Acknowledgments:** Our thanks are due to the CAS Big Earth data platform for providing the impervious surface dataset in the study area.

**Conflicts of Interest:** The authors declare no conflict of interest.

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
