# Peer review of "Big Data Supported the Identification of Urban Land Efficiency in Eurasia by Indicator SDG 11.3.1"

_ijgi, doi:10.3390/ijgi10020064_

Round 1

Reviewer 1 Report

The paper is very interesting, but it misses some key literature references.

In the INTRODUCTION, the authors should add, after line 41, the references by Anderson et al. (2017) and by Scott and Rajabifard ( 2017).

In Section 2.2 the authors have missed some recent important references such as Shao et al.(2020) and Huang and Wang (2020)

In section 2.3, the authors should cite additional literature such as recent paper by Trinder and Liu (2020)

See below: 

John Trinder & Qingxiang Liu (2020) Assessing environmental impacts of urban growth using remote sensing, Geo-spatial Information Science, 23:1, 20-39, DOI: 10.1080/10095020.2019.1710438

Zhenfeng Shao, Neema S. Sumari, Aleksei Portnov, Fanan Ujoh, Walter Musakwa & Paulo J. Mandela (2020) Urban sprawl and its impact on sustainable urban development: a combination of remote sensing and social media data, Geo-spatial Information Science, DOI: 10.1080/10095020.2020.1787800

Greg Scott & Abbas Rajabifard (2017) Sustainable development and geospatial information: a strategic framework for integrating a global policy agenda into national geospatial capabilities, Geo-spatial Information Science, 20:2, 59-76, DOI: 10.1080/10095020.2017.1325594

Bo Huang & Jionghua Wang (2020) Big spatial data for urban and environmental sustainability, Geo-spatial Information Science, 23:2, 125-140, DOI: 10.1080/10095020.2020.1754138

Huadong Guo, Jie Liu, Yubao Qiu, Massimo Menenti, Fang Chen, Paul F. Uhlir, Li Zhang, John van Genderen, Dong Liang, Ishwaran Natarajan, Lanwei Zhu & Jiuliang Liu (2018) The Digital Belt and Road program in support of regional sustainability, International Journal of Digital Earth, 11:7, 657-669, DOI: 10.1080/17538947.2018.1471790

Author Response

Dear reviewer,

First of all, I would like to thank you for your critical comments on our manuscript. According to your comments, we carefully revised our manuscript. Please see detail below and the revised version.

Reviewer#1:

The paper is very interesting, but it misses some key literature references.

Reply: Thanks for your kind help. We have read carefully the literatures you mentioned and added them, please see detail in the sections of reference.

In the INTRODUCTION, the authors should add, after line 41, the references by Anderson et al. (2017) and by Scott and Rajabifard ( 2017).

Reply: We have added these two literatures after line 41 with reference numbers 8,9.

In Section 2.2 the authors have missed some recent important references such as Shao et al.(2020) and Huang and Wang (2020)

Reply: We have added these two literatures along with Guo et al. (2018) at the beginning of section 2.2 with reference numbers 31,32 and 35.

In section 2.3, the authors should cite additional literature such as recent paper by Trinder and Liu (2020)

Reply: We have added this work in section 2.3 with reference number 36.

Reviewer 2 Report

In its present form, this work is a mere descriptive analysis of the relation between population growth rate and land consumption rate for around 1000 cities part of the so-called "Belt and Road" initiative. In this sense, the title is misleading, as the paper is not dealing with big data at all. The authors took well-known (and debatable) formulations and using already available data, and they calculated the LCRPGR indicator for all the selected cities. They cluster their results into three categories using the mean value of the LCRPGR as a threshold between clusters. And that's it. All the after discussion verse around describing the clusters without any real analysis of the reasons behind such observed values. If something, they should further discuss the actual LCRPGR and its individual components. Why Ln? What are the units of each formulation? Is there a common factor among cities in each cluster? 

And about the abnormal values, it is not entirely clear why they should be removed. At least not with the explanation provided. This will be clearer if the authors spend more time discussing the validity of the LCRPGR.

In general, this is not a robust work. There is no clear research question on it. My advice is that, if the authors intend to resubmit, it would be not just a matter of address one or two points, but a whole reinking of the work and what is the actual message. 

Some miscellaneous comments:

Line 136 reads 1900 instead of 1990.

The Data and Data Processing section is poorly explained, and the reader needs to understand how this "impervious surface" is derived. 

Figure 1 is not good at all. There are some extra legends (like coastlines), others are wrong (National boundaries is label as polygon when is a line), the point data is messy. There are plenty of techniques to plot thousands of points in a map clearly. 

Author Response

Thanks for your valuable comments, please see detail in the attached file.

Reviewer 3 Report

The paper is very well structured. In detail, describe carried research and its results. It has aim to calculate the UN Sustainable Development Goal indicator 11.3.1 values and examine the land use efficiencies in two measurement periods. It is based to the known methodology, which is tested through large number of cities.

However, the paper should reconsider some minor technical improvements:

  • The reference 21 is not nominated in text
  • In the paper is often used “study” which should be changed with word “paper”
  • In the Abstract are some numbers in bracket but it is not clear which meaning they have. Are they connection with the equations? If it is so, are they good represented?
  • Name of the figure should be on the same page with the figure (see figure 4)
  • Name of the chapter should be fallowed with at least one sentence on the same page (see location of title 5 name)
  • After every chapter name should be some entrance sentence. It is not appropriate to have chapter title follow with chapter subtitle (see chapter 2)
  • The paper should be written in third face of singular. So first face of singular is not appropriate (should not use expression “we”, “our” etc.)

Author Response

Dear reviewer,

First of all, I would like to thank you for your critical comments on our manuscript. According to your comments, we carefully revised our manuscript. Please see detail below and the revised version.

Reviewer#3:

The paper is very well structured. In detail, describe carried research and its results. It has aim to calculate the UN Sustainable Development Goal indicator 11.3.1 values and examine the land use efficiencies in two measurement periods. It is based to the known methodology, which is tested through large number of cities.

However, the paper should reconsider some minor technical improvements:

The reference 21 is not nominated in text

Reply:Thanks, we have nominated the reference 21 as new reference number 23 in the revised manuscript.

In the paper is often used “study” which should be changed with word “paper”

Reply:Thanks, based on your suggestions, we have substituted some of the “study” with “paper” or “work” by proofreading the revised manuscript. Below is the detail:

Line numbers: 93, 254, 287

In the Abstract are some numbers in bracket but it is not clear which meaning they have. Are they connection with the equations? If it is so, are they good represented?

Reply:We have removed the numbers in bracket in the Abstract of the revised version. In addition, we have reorganized the section of Abstract.

Name of the figure should be on the same page with the figure (see figure 4)

Reply: thanks, we have kept a figure in a same page in the revised version.

Name of the chapter should be fallowed with at least one sentence on the same page (see location of title 5 name)

Reply: thanks, we have kept at least one sentence on the same page followed by the name of a chapter in the revised version.

After every chapter name should be some entrance sentence. It is not appropriate to have chapter title follow with chapter subtitle (see chapter 2)

Reply: thanks, we have added the entrance sentence after every chapter in the revised version. A sentence was added after chapter 2 and listed below:

In Section 2.1, we briefly introduce the study site, and the city samples and distributions in the study area. In Section 2.2, we present the data sources and processing procedures, and section 2.3 is the method that was applied in this study to estimate SDG 11.3.1 according to the UN metadata.

The paper should be written in third face of singular. So first face of singular is not appropriate (should not use expression “we”, “our” etc.)

Reply: thanks, we have replaced the first face with third face of singular in the revised version. We appreciate your kind help in how to write English well.

Reviewer 4 Report

Review on the paper entitled “Big data supported analysis of urban land efficiency along the Belt and Road by indicator SDG 11.3.1”.

This paper aims to calculate LCRPGRs of cities located in different geographical regions in two time periods. For me this paper seems to be a technical report on how the SDG 11.3.1 indicator works rather than a research paper. In my opinion, the authors should make more effort to convert this study to a decent research paper. Let me present my observations in the followings:

First of all, the paper claims that “Europe and Japan have the lowest PGR and LCR” and “South and Central Asia have the lowest values of LCRPGR”. Have these different values been produced by the different urbanization level exclusively? I think it is more complicated. For example, they can be caused by the difference between the strictness of regulations of land use in countries located in those regions. In many European countries, it is strictly prohibited to convert natural lands into built-up areas regardless of the magnitude of the population growth (of course most European countries experience population decline). I think the authors should provide a more straightforward explanation regarding the LCRPGR in different regions.

Second, I really do not understand why the authors involved the “Belt and Road” into this paper. The Belt and Road initiative is an infrastructure development strategy of China of which geographic scope covers almost all countries in Asia and the Middle East. To my best knowledge not every European country is part of Belt and Road plan, yet the authors investigated the entire region. That is, their analysis simply focuses on the investigation of LCRPGR in some large geographical regions of Eurasia including the Middle East. The analysis has nothing to do with the Belt and Road initiative.

In line 75, there is a rather strange expression: sub-city. Do the authors want to refer by this term to suburban settlements? How should we understand “sub-cities”?

In line 133, there is an equation. I think, the third term of the equation (following the second equation sign) is not necessary to be indicated here, as it is simply the repetition of equations 1 and 2.

The authors define some LCRPGR values to be “abnormal”. Then they exclude 24.19 percent of the cities from the analysis (88 out of the 421 and 143 out of the 534 cities were excluded). I think it is daring to say that for almost one-fourth of the cities being involved in the analysis the LCRPGR value is “abnormal”.

As far as I am aware, most researchers, even such organizations as the United Nations, define megacities as cities with more than 10 million inhabitants. I am not sure that cities with 5 million inhabitants are considered as megacities (lines 153-155). Of course, I can accept if the authors decide to employ this classification.

What is the meaning of Urb1990 and Urb2000 in Table 1? There is no explanation and measure here but an abbreviation.

In Table 2, the authors intend to present some data ranges of the LCRPGR values. I am bit confused at this point because in the Table there are only some numbers in brackets, for example [1,5], that, in my opinion, do not refer to ranges. If the authors use this formula [1-5] then I will be aware that they refer to a range of values. In addition, if the LCRPGR value is 0, then what will it suggest for us: “rapid urban population growth” or “urban shrinking”? The problem is that some thresholds belong to more categories at the same time.

I cannot understand the meaning of this sentence (lines 173-174): “We can see from PGR that Southeast Asia, Europe and Japan has the biggest and smallest population growth rate.” Do they have the biggest or the smallest population growth rate?

Also here the following sentence appears (lines 176-177): “While, Europe and Japan is the only one region with an increased population growth rate from periods 1990-2000 to 2000-2015 even though the population growth rate is still lower than other areas.” First of all, Europe and Japan are two regions being located far away from each other. Second, the authors should be careful when drawing conclusion on countries population change based on the population trend of some cities. Third, please check the meaning of this sentence. How did “only” Europe and Japan produce population growth if other regions did the same?  

In lines 210-213, the followings can be read: “This indicates that the urban lands in large mega and mega cities are denser than that of large, medium and small size cities indicating relatively higher urban land efficiency in large mega and mega cities, which means that the land use efficiency has become higher as the process of urbanization intensified.” It is worth noting that as per the practice of the United Nations, megacities across the world but with the exception of China are created by merging central cities and other settlements (cities, towns, villages, etc.) located in their hinterlands. For example, according to UN data, Paris has more than 11 million inhabitants in 2020, that is, based on this value, Paris is considered to be a megacity. However, in reality, Paris has about only 2.2 million inhabitants, whereas the UN considers the entire Île-de-France region with about 1,700 settlements to be part of Paris. Of course, the centre city (i.e., the City of Paris) produces a high urban land efficiency but, I suppose, it is not true for the remaining parts of the megacity.

Author Response

Dear reviewer,

First of all, I would like to thank you for your critical comments on our manuscript. According to your comments, we carefully revised our manuscript. Please see detail in the attached file and the revised version.

Round 2

Reviewer 2 Report

The authors performed a major rewritten effort to improve their manuscript and took into consideration all my previous concerns. I don't have further objections for its publication.

Author Response

Dear reviewer,

Thanks for your valuable comments in the first round reivew which made our mansucript improved a lot! Thanks again!

Reviewer 4 Report

Review on the paper entitled “Big data supported identification of urban land efficiency in Eurasia by indicator SDG 11.3.1”.

I appreciate the work of the Authors they did to improve the quality of the paper.

I have only one remark. I think that Europe and Japan have been classified into the same category not because of the fact that both of them are developed countries (this is true for the US and Australia as well) but because the population of both Europe and Japan have been decreasing for a while. This is a quite unique feature of these regions. The Authors may investigate this issue if they are curious but, as a matter of fact, it does not have effect on the scientific content of the paper.

Author Response

Dear Reviewer,

I truly don't konw the reason why Eurpean and Japan is put into tegother as a unique region in UN report. Thanks for your remark, I know the real reason now. Thanks again.

Our work can not be published without your valuable comments in the first review round. We appreacite your help very much!

Best wishes

Guoyin Cai